# Sourdough Bread with Different Fermentation Times: A Randomized Clinical Trial in Subjects with Metabolic Syndrome

**DOI:** 10.3390/nu16152380

**Published:** 2024-07-23

**Authors:** Karla Alejandra Pérez-Vega, Albert Sanllorente, María-Dolores Zomeño, Ana Quindós, Júlia Muñoz-Martínez, Mireia Malcampo, Ana Aldea-Perona, Álvaro Hernáez, Aleix Lluansí, Marc Llirós, Isidre Elias, Núria Elias-Masiques, Xavier Aldeguer, Daniel Muñoz, Sònia Gaixas, Gemma Blanchart, Helmut Schröder, Javier Hernando-Redondo, Nerea Carrón, Pedro González-Torres, Valentini Konstantinidou, Montserrat Fitó, Olga Castañer

**Affiliations:** 1Hospital del Mar Research Institute, 08003 Barcelona, Spain; kperez@researchmar.net (K.A.P.-V.); asanllorente.apms.ics@gencat.cat (A.S.); mariadoloreszf@blanquerna.url.edu (M.-D.Z.); aquindos@psmar.cat (A.Q.); juliamm1@blanquerna.url.edu (J.M.-M.); mmalcampo@researchmar.net (M.M.); aaldea@researchmar.net (A.A.-P.); alvarohc1@blanquerna.url.edu (Á.H.); dmunoz@researchmar.net (D.M.); sgaixas@researchmar.net (S.G.); gblanchart@researchmar.net (G.B.); hschroder@researchmar.net (H.S.); jhernando1@researchmar.net (J.H.-R.); ocastaner@researchmar.net (O.C.); 2Consorcio Centro de Investigación Biomédica En Red (CIBER), M.P. Fisiopatología de la Obesidad y Nutrición (CIBEROBN), Instituto de Salud Carlos III, 28029 Madrid, Spain; 3PhD Program in Food Science and Nutrition, Universitat de Barcelona, 08028 Barcelona, Spain; 4Unitat de Suport a la Recerca Metropolitana Sud, Fundació Institut Universitari per a la Recerca a l’Atenció Primària de Salut Jordi Gol i Gurina (IDIAPJGol), 08907 Hospitalet de Llobregat, Spain; 5Direcció d’Atenció Primària Metropolitana Sud, Institut Català de la Salut, 08907 Hospitalet de Llobregat, Spain; 6Blanquerna School of Health Sciences, Universitat Ramon Llull, 08025 Barcelona, Spain; 7Global Research on Wellbeing Research Group (GRoW), Facultat de Ciències de la Salut Blanquerna, Universitat Ramon Llull, 08025 Barcelona, Spain; 8Research Group on Pedagogy, Society and Innovation with ICT Support (PSITIC), Facultat de Psicologia, Ciències de l’Educació i l’Esport Blanquerna, Universitat Ramon Llull, 08022 Barcelona, Spain; 9Digestive Diseases and Microbiota Group, Institut d’Investigació Biomèdica de Girona, 17190 Salt, Spain; aleix.lluansi@sjd.es (A.L.); marc.lliros@uvic.cat (M.L.); xaldeguer@idibgi.org (X.A.); 10Institut de Recerca Sant Joan de Déu (IRSJD), Hospital Sant Joan de Déu, 08950 Barcelona, Spain; 11Bioinformatics and Bioimaging (BI-SQUARED) Research Group, Biosciences Department, Faculty of Sciences, Technology and Engineerings Universitat de Vic—Universitat Central de Catalunya, 08500 Vic, Spain; 12Elias–Boulanger S.L., 08340 Vilassar de Mar, Spain; ielias@lamethode.eu (I.E.);; 13GoodGut S.L., 17003 Girona, Spain; 14Digestive Service, Hospital Universitari de Girona Dr. Josep Trueta, 17007 Girona, Spain; 15Consorcio Centro de Investigación Biomédica En Red (CIBER), M.P. Epidemiología y Salud Pública (CIBERESP), Instituto de Salud Carlos III, 28029 Madrid, Spain; 16Microomics Systems S.L., 08041 Barcelona, Spain; nerea.carron@microomics.com (N.C.); pedro.gonzalez@microomics.com (P.G.-T.); 17Medoliali S.L. (DNANUTRICOACH®), 08006 Barcelona, Spain; valentini@dnanutricoach.com; 18Faculty of Health Sciences, Universitat Oberta de Catalunya (Open University of Catalonia, UOC), 08018 Barcelona, Spain

**Keywords:** sourdough bread, metabolic syndrome, sICAM, PAI-1, blood pressure

## Abstract

The Mediterranean diet, featuring sourdough bread, shows promise in managing metabolic syndrome. This study explored the effects of two sourdough breads, with differing fermentation times but similar nutritional profiles, on inflammation, satiety, and gut microbiota composition in adults with metabolic syndrome. In a double-blind clinical trial, participants were randomized to consume either Elias Boulanger^®^ long-fermentation (48 h) sourdough bread (EBLong) or Elias Boulanger^®^ short-fermentation (2 h) sourdough bread (EBShort) over a two-month period. We assessed clinical parameters, inflammatory biomarkers, satiety-related hormones, and the richness and abundance of gut microbiota at baseline and follow-up. The participants included 31 individuals (mean age, 67, 51.6% female). EBShort was associated with reduced levels of soluble intercellular adhesion molecule (sICAM), and all participants, regardless of the intervention, exhibited a decrease in sICAM and diastolic pressure from baseline (*p* < 0.017). At follow-up, plasminogen activator inhibitor-1 (PAI-1) levels were lower in EBShort (−744 pg/mL; 95%CI: −282 to −1210 pg/mL) compared to EBLong. No differences in microbiota richness or abundance were observed. EBShort bread was effective in reducing some inflammation markers. The consumption of sourdough bread may offer potential benefits in reducing inflammation markers in individuals with metabolic syndrome; however, longer fermentation times did not show additional benefits.

## 1. Introduction

Metabolic syndrome, a cluster of closely interrelated metabolic disorders, is strongly associated with cardiovascular disease incidence and the risk of developing type 2 diabetes mellitus [1,2]. The risk factors for metabolic syndrome include abdominal obesity, high triglyceride levels, low concentrations of high-density lipoprotein cholesterol (HDLc), elevated blood pressure, insulin resistance, and/or hyperglycemia [3]. The adoption of healthy lifestyle habits such as an adequate diet, weight management, regular physical activity, and avoiding smoking can reduce cardiometabolic risk [4]. However, an integrated approach to prevention is essential. This could involve policies that facilitate access to healthy foods and physical activity, as well as innovations within the food industry.

In our context, the Mediterranean diet (MedDiet), traditionally characterized by the triad, with the special importance of bread [5], has proven effective in cardiovascular disease prevention and in improving risk factors associated with metabolic syndrome [6,7]. Bread has been a crucial dietary component for Mediterranean populations since ancient times [8]. Sourdough bread, dating back centuries, is considered a cultural heritage. It contains microorganisms such as yeast and lactic acid bacteria. The dough, made from wheat and/or rye flour mixed with water, is left to naturally ferment at room temperature, enhancing its microbial content [9]. These microorganisms produce the gases and acidity that give the distinctive aroma, texture, flavor, and qualities to sourdough bread [10]. These characteristics arise from the prolonged fermentation times, unlike commercial methods that involve shorter periods and higher temperatures [11]. Finally, sourdough bread, which contains probiotics, prebiotics, and by-products from microbial metabolism, has demonstrated potential in addressing chronic non-communicable diseases [12,13]. However, research on this topic is limited, and further studies are necessary to confirm its potential benefits for metabolic syndrome and cardiovascular health.

The present study aimed to evaluate the effects of two sourdough breads, one with long (48 h) and one with short (2 h) fermentation, on clinical parameters, inflammatory biomarkers, and satiety-related hormones in individuals with metabolic syndrome. Additionally, this study aimed to characterize the gut microbiota as a secondary objective. Our aim was to demonstrate that long-fermentation bread could potentially yield superior results.

## 2. Materials and Methods

### 2.1. Study Design and Population

This is a randomized, double-blind clinical trial with two parallel groups.

Eligible participants were community-dwelling adults who met at least three components of metabolic syndrome according to the updated harmonized criteria of the joint statement from the International Diabetes Federation, the National Heart, Lung and Blood Institute, and the American Heart Association: hypertriglyceridemia [≥150 mg/dL (≥1.7 mmol/L)] or drug treatment for elevated triglycerides; low concentrations of HDLc [<50 mg/dL (<1.3 mmol/L) and <40 mg/dL (<1.03 mmol/L) in women and men, respectively] or drug treatment for low HDLc; elevated blood pressure (systolic ≥ 130 mmHg and/or diastolic ≥85 mmHg) or being treated for hypertension; high fasting plasma glucose [≥100 mg/dL (≥5.5 mmol/L)] or drug treatment; and elevated waist circumference for European individuals (≥88 cm in women and ≥102 cm in men) [3]. Exclusion criteria included the following: (a) the use of antibiotic, prebiotic, and/or probiotic supplements in the 3 months prior to the start of the study; (b) celiac disease; (c) inflammatory bowel disease; (d) a history of bowel resection; (e) alcoholism and/or any other drug dependence; (f) the use of non-steroidal anti-inflammatory drugs; (g) the use of immunosuppressants, antibiotics, and proton pump inhibitors; (h) any disease or condition preventing compliance with the study protocol; and (i) inability to provide informed consent.

Participants were assigned to one of two sourdough bread (Elias Boulanger^®^, Dresden, Germany) intervention arms: (a) Elias Boulanger^®^ long-fermentation bread (EBLong) or (b) Elias Boulanger^®^ short-fermentation bread (EBShort). The volunteers were randomized between the two intervention arms using a random sequence generated by a computer program (Cardiovascular Epidemiology and Genetics Group, EGEC, Hospital del Mar Research Institute, Barcelona, Spain). The double-blind protocol was maintained until the analysis of the results. No recommendations regarding diet, physical activity, or lifestyle were provided. Participants were instructed to replace their usual bread with the intervention bread while maintaining the same quantity consumed over a two-month period. Participants were instructed to collect sliced bread from their chosen bakeries once a week or every two weeks (from 15 bakeries distributed throughout the province of Barcelona) and freeze it. This trial was conducted at the Hospital del Mar Research Institute in Barcelona, Spain.

### 2.2. Ethical Aspects

The study protocol complied with the Declaration of Helsinki for Medical Research involving Human Subjects. It was reviewed and approved by the Clinical Research Ethics Committee of the Parc de Salut Mar Barcelona consortium (CEIC-Parc de Salut Mar). Protocol code: 2017/7752/I. All participants read and signed an informed consent form before enrollment in the study.

### 2.3. Bread Composition and Fermentation Process

First, a starter was prepared by blending equal parts by weight of whole-grain wheat flour (T110 flour, Triticum dicoccoides; Moulin de Colagne^®^, Bourgs sur Colagne, France) and water, which rested for 24 h at room temperature. On day 5, a mixture of infusions, dairy products, and fruit (following the baker’s own recipe) was added and left at room temperature. The final starter had the following characteristics: a temperature of 34 °C, a pH value of 6.68, and a total titratable acidity of 13.73 mL. Sourdough was created by combining the starter (30%), water (50%), and whole-grain wheat flour milled through a stone mill (70%, T110 flour, T. dicoccoides; Moulin de Colagne^®^, Bourgs-sur-Colagne, France). Subsequently, a process known as feeding or back-slopping was carried out, involving the use of the initial mixture to ferment a new blend of water and flour [14], at regular intervals of every 12–24 h for 3 days, followed by fermentation for 4 days at 10−15 °C. This procedure aimed to stimulate microbial fermentation and promote the propagation of sourdough [15]. The details of the high-quality sourdough preparation can be found elsewhere [16].

EBLong was formulated with sourdough (wt/wt; 30% flour basis), whole-grain flour (*Triticum aestivum*), water (wt/wt; 80% flour basis), dry baking yeast (*Saccharomyces cerevisiae*, wt/wt; <0.5% flour basis; Lesaffre (Hirondelle^®^)), and salt (wt/wt; 1.1% flour basis; Guerande^®^). The bread was then fermented in a fermentation chamber (Eurifours^®^, Gommegnies, France) for 48 h: 46 h of maturation at 4–6 °C and 2 h of dough development at 28 °C.

EBShort was composed of sourdough (wt/wt; 10% flour basis), refined wheat flour (Farinera Corominas^®^, Banyoles, Spain) (being a blend of white bread since the starter contained whole-grain flour), water (wt/wt 60% flour basis), yeast (*S. cerevisiae*, 0.010 g/kg of flour; Lesaffre (Hirondelle^®^)), salt (wt/wt 1.2% flour basis, Sal Costa^®^), enzymes(α-amylase, endoxylanasse, amyloglucosidanassa; Uniplus^®^, Manassas, VA, USA), wheat gluten (0.010 g/kg of flour; Uniplus^®^), xanthan gum (0.005 g/kg of flour), and additive components: emulsifier E471, antioxidant E-300 (Uniplus^®^). It was fermented in a chamber (Eurifours^®^) for 2 h at 28 °C.

Both breads were baked after the fermentation process in a Eurifours^®^ oven at 200 °C for 90 min. The nutritional content of both breads can be found in Appendix A.

### 2.4. General and Lifestyle Data

The following variables were recorded at baseline and follow-up: (a) a questionnaire for adherence to the MedDiet [17]; (b) a three-day food record collected during the three days prior to each visit (including one weekend day), which was further translated into nutrient values using the Pro-PCN software (Barcelona, Spain) [18]; and (c) an abbreviated questionnaire from the Minnesota Leisure Time Physical Activity Questionnaire [19].

### 2.5. Anthropometric and Exploration Data

The following variables were measured at baseline and follow-up: participants’ weight was recorded in kilograms to one decimal point using a high-quality electronic scale, rounded to the nearest 100 g, with individuals wearing light clothing and no shoes, jackets, or coats. Height was measured in centimeters with a stadiometer accurate to 1 cm. Waist circumference was measured at the midpoint between the last rib and the iliac crest, on expiration, at the most prominent point of the trochanter. Body mass index (BMI) was derived by dividing weight (in kilograms) by height squared (in meters squared). Blood pressure was assessed while participants were seated with their backs and arms supported to ensure the cuff was at heart level, after refraining from smoking or consuming caffeine for 30 min. Measurements were taken on both arms using appropriate cuffs, with the arm displaying the higher mean diastolic blood pressure selected for subsequent measurements throughout the study. If the first two readings differed by more than 5 mmHg, additional readings were taken and averaged.

### 2.6. Laboratory Analysis

The following parameters were analyzed in fasting ethylenediaminetetraacetic acid (EDTA) plasma at baseline and after a two-month follow-up: glucose (Glucose HK CP, Horiba ABX, Montpellier, France), triglycerides (Triglycerides CP, Horiba ABX), total cholesterol (Cholesterol CP, Horiba ABX), and HDLc (HDLc Direct CP, Horiba ABX, Deurne, The Netherlands) were measured in an autoanalyzer ABX Pentra (Horiba ABX SAS, Spain). The HOMA index was calculated as (glucose × insulin)/405. We calculated LDLc with the Friedewald formula only when triglycerides were <300 mg/dL; higher values (≥300 mg) resulted in a missing value for LDLc. The following inflammation markers were analyzed: interleukin 6 and 8 (IL6 and IL8), tumor necrosis factor alpha (TNF-α) (Bio-Plex Cytokine 8-plex, Bio-Rad, Hercules, CA, USA), soluble intercellular adhesion molecule (sICAM) (Bio-Plex Cytokine 2-plex, Bio-Rad), and plasminogen activator inhibitor-1 (PAI-1) (Bio-Plex Pro Human Diabetes 10-plex, Bio-Rad). Additionally, satiety-related hormones were analyzed: insulin, C-peptide, ghrelin, leptin, glycoprotein 1 (GLP-1), glucagon, resistin, and visfatin (Bio-Plex Pro Human Diabetes 10-plex, Bio-Rad). These analyses employed Luminex^®^ xMAP^®^ technology in a BioPlex system (Bio-Rad, Hercules, CA, USA). The lipopolysaccharide binding protein (LBP) (Human LBP, Hycult Biotech, Uden, The Netherland) was measured with an ELISA kit. All samples, both baseline and follow-up, were analyzed in the same run on the same day. The intra- and inter-assay coefficients of variation of the inflammatory and satiety-related biomarkers are in Appendix A.

### 2.7. Intestinal Microbiota Analysis

Participants collected fecal samples at home in sterile containers provided for this purpose. They were instructed to keep their samples frozen until delivering them to the study staff the following day. The samples were sent to the Girona Biomedical Research Institute (IdIBGi, Salt, Spain) on dry ice to maintain the cold chain.

Genomic DNA was extracted from approximately 0.25 g of feces using commercial methods (NucleoSpin Soil kit, Macherey-Nagel^®^, Hoerdt, France). The quality and quantity of the extracted nucleic acids were measured by a Nanodrop ND 2000 UV-Vis spectrophotometer (Thermo Fisher Scientific, DE, USA) and Qubit^®^ (Thermo Fisher Scientific, DE, USA) according to the manufacturers’ instructions. The region corresponding to the variable V3-V4 region of the 16S rRNA gene was determined with specific primers (341F/806R, [20]) and Illumina technology HiSeq 2000 using paired-end reads (generating 300 bp sequences). Briefly, amplification was performed after 25 PCR cycles. In this procedure, positive (CM) and negative (CN) controls were used to ensure quality control. The positive control is a Mock Community DNA (Zymobiomics Microbial Community DNA) control, and it was processed the same way as the samples.

### 2.8. Bioinformatic Analysis

Raw demultiplexed forward reads were processed using the following methods and pipelines as implemented in QIIME2 version 2020.11 with default parameters unless stated [21]. DADA2 was used for quality filtering, denoising, and amplicon sequence variant calling (ASV, i.e., phylotypes) using qiime dada2 denoise-single method [22]. Q16 was used as a quality threshold to define read sizes for trimming (parameter: -- p-trunc-len). Reads were truncated at the position when the 75th percentile Phred score fell below Q16: 198 bp. After quality filtering steps, the average sample size was 64,907 reads (min: 37,026 reads, max: 153,151 reads), and 1896 phylotypes were detected. Amplicon sequence variants (ASVs) were aligned using the qiime alignment mafft method [23]. The alignment was used to create a tree and to calculate phylogenetic relations between ASVs using the qiime phylogeny fasttree method [24]. ASV tables were subsampled without replacement in order to even sample sizes for diversity analysis (alpha and beta diversities) using the qiime diversity core-metrics-phylogenetic pipeline. The smallest sample size was chosen for subsampling (i.e., 37,000 reads). Subsequently, reads were clustered into 1896 operational taxonomic units (OTUs). The following alpha diversity metrics were calculated: observed ASV number (i.e., richness) and Pielou’s evenness index. For the alpha and beta diversity analyses, ASVs were used without considering the taxonomic level of each ASV. Weighted Unifrac distances were calculated to compare community structure [25].

The taxonomic assignment of phylotypes was performed using a Bayesian Classifier [26] trained with Silva database version 138 (99% OTUs full-length sequences) [27], using the qiime feature-classifier classify-sklearn method [28].

### 2.9. Sample Size

Considering a two-sided type I error of 0.05, a power of ≥80%, and a standard deviation of PAI-1 (840 pg/mL) obtained from a previous study, a sample size of 31 participants per group would allow for the detection of a difference of 600 pg/mL. Additionally, a sample size of 30 subjects provides sufficient power to detect a change of 866 pg/mL.

### 2.10. Statistical Analysis

This study described categorical variables by proportions, normally distributed continuous variables by means and standard deviations (SDs), and non-normally distributed continuous variables by medians (1st–3rd quartile). Changes relative to pre-intervention values were assessed in both groups separately and together by paired t-tests for normally distributed continuous variables and Wilcoxon signed-rank tests for non-normally distributed variables. Multivariable linear regressions were conducted to explore whether there were differences in the follow-up values in the EBLong group relative to the EBShort group. The models were adjusted for baseline levels of each parameter (continuous), age (continuous), sex, BMI (continuous), and MedDiet adherence (continuous).

Given that our variables showed a high correlation, we conducted Horn’s parallel analysis, a factor analysis tool, before correcting for multiple comparisons. This analysis identified the number of independent components or factors that best described our variables [29]. We identified three significant factors in our data. We used these factors to apply the Bonferroni correction, considering values significant when the *p*-value was less than 0.05/3 (<0.017). Analyses were performed using R Software version 4.3.1 [30].

Metagenome statistical analyses were conducted using Generalized Linear Mixed Models (GLMMs). Alpha diversity comparisons were performed using R package NBZIMM version 1.0 [31] for richness and the R package betareg version 3.1-4 [32] for evenness. Beta diversity distance matrices were used to perform a principal coordinates analysis; the significance of groups in the community structure was tested using PERMANOVA, a non-parametric test. The differential abundance of taxa was tested using Negative Binomial Generalized Linear Mixed Models using the R package NBZIMM [31]. P-values were adjusted using the false discovery rate (FDR). A significant threshold was set at 0.05.

## 3. Results

### 3.1. Study Population

Participants were recruited between July 2019 and February 2020. A total of 292 individuals were contacted, and finally, 61 were enrolled (55.7% female) (Figure 1). Of these, 31 were randomly assigned to the EBLong intervention and 30 to the EBShort intervention. Due to the COVID-19 pandemic, only 31 participants finished the study, resulting in 13 participants in the EBShort group and 18 in the EBLong group. The participants were older adults, with a mean age of 66.7 years, and 51.6% were female. They exhibited a high prevalence of metabolic risk factors. The only variable in which intergroup differences were observed in baseline values was BMI. No differences in diet according to the Adherence to MedDiet questionnaire, kcal intake, and physical activity performance at baseline were found (Table 1).

### 3.2. Dietetic Assessment

Participants in both groups had similar diets before and during the last week of the study. No differences were found in the quantity of energy or macronutrient intake. The intake of some of the main nutrients recorded by the three-day food records can be found in Appendix A.

### 3.3. Clinical Parameters, Inflammatory Biomarkers, and Satiety-Related Hormones

Irrespective of the intervention, after two months, all participants had a decrease in sICAM and diastolic pressure (*p* < 0.017) (Table 2). While we observed a decrease in sICAM and PAI-1 levels in the EBShort group (Table 2), a trend towards a decrease in diastolic pressure was determined in the EBLong group (*p* = 0.020) (Table 2).

No statistically significant differences were observed between groups in the follow-up values of these variables in either the non-adjusted model or adjusted model, except for PAI-1 in the EBLong group compared to EBShort (744 pg/mL; 95%CI: −282 to −1210 pg/mL) (Table 3).

We did not observe any statistically significant differences in other clinical parameters, inflammation biomarkers, or satiety-related hormones after the intervention or between the intervention groups.

### 3.4. Microbiota Characterization

Alpha diversity analysis revealed no discernible differences in richness and evenness within or between the intervention groups during the follow-up period (Appendix A). Beta diversity showed no significant distinctions between the treatment groups throughout the study period (PERMANOVA R2 = 0.0072, *p* = 0.389) (Appendix A). Figure 2 illustrates the relative frequencies of the most abundant microbial phyla and family. There were no differences in the baseline relative abundance of phyla between the groups. After the intervention with EBLong bread, there was a significant reduction in Synergistota abundance relative to baseline (*p* < 0.001). Additionally, at the follow-up assessment, the EBLong group exhibited a lower abundance of Synergistota compared to the EBShort group, although its abundance did not exceed 1% (see Appendix A). At the family level, after the two-month intervention, the EBLong group showed a reduction in Enterobacteriaceae and Synergistaceae. In contrast, the EBShort group exhibited a decrease in Oscillospiraceae and an increase in Prevotellaceae. The abundance of Oscillospiraceae differed significantly between the groups at follow-up; however, this difference was also observed at baseline.

## 4. Discussion

While the main components of the MedDiet have been extensively studied, the role of bread in managing metabolic syndrome is less understood. This study aimed to evaluate the impact of high-quality sourdough bread on individuals with metabolic syndrome by examining two variants with different fermentation times. Over two months, participants showed minimal changes in inflammation markers. However, all participants had reduced sICAM levels and diastolic blood pressure after consuming either type of bread. The EBShort bread led to significant decreases in sICAM and PAI-1 levels compared to baseline. At follow-up, only the EBShort group showed significantly lower PAI-1 levels compared to the EBLong group. Unexpectedly, no further changes were observed in the bread with longer fermentation. Microbiota analysis indicated stable gut microbial communities with no significant changes in diversity or abundance following the intervention.

Inflammation plays an important role in the development and progression of metabolic syndrome [33]. Therefore, seeking measures to mitigate inflammation holds potential for improving metabolic and cardiovascular health outcomes. Regarding inflammatory biomarkers, no significant differences in cytokines were observed, consistent with previous findings [34]. Nevertheless, participants who replaced their regular bread with sourdough (either short or long fermentation) for two months showed lower levels of sICAM, an adhesion molecule essential in atherosclerosis development [35]. Comparable outcomes were found in a study involving a 5-week intervention with a prebiotic antioxidant bread comprising wheat–rye bread, tomato paste, green tea powder, and herbs [36]. Seidel et al. also reported a significant reduction in ICAM-1 levels among non-smoking adults following the intervention, despite their bread lacking sourdough. Their bread was similarly enriched with infusions, dairy, and fruit, resulting in a flavonoid-rich, pre- and probiotic bread that may confer this effect. An in vitro study suggests that some flavonoids could inhibit ICAM-1 expression [37], while a clinical trial demonstrated reduced ICAM-1 expression after consuming a probiotic sausage containing *Lactobacillus paracasei* [38]. Given this study’s limited number of participants and dietary variability in flavonoids and probiotics, further evidence is needed to establish any causal mechanisms behind these findings. PAI-1 is a primary inhibitor of fibrinolysis, playing a critical role in the development of thrombosis, atherosclerosis, and cardiovascular risk [39]. Contrary to our hypothesis that a higher proportion of sourdough in bread and longer fermentation would reduce inflammation, we observed lower PAI-1 levels in the EBShort group compared to the EBLong intervention at follow-up. Since our objective was to compare different sourdough breads, distinguishing between interventions is challenging without a poor-quality control bread for reference. In another study, researchers compared whole-grain wheat sourdough bread to refined white bread in normoglycemic–normoinsulinemic and hyperglycemic–hyperinsulinemic patients and found no differences in PAI-1 levels after six weeks of consumption [40]. The different sample sizes of our intervention groups, the greater proportion of diabetics, differences in BMI (being greater in EBLong), and the use of different breads may explain discrepancies in the findings presented here.

Regarding clinical parameters, after consuming sourdough bread for two months, we observed a decrease in diastolic pressure. However, this effect was not evident when comparing each individual intervention at follow-up. These results may be influenced by the small sample size. Nonetheless, these data should be interpreted cautiously due to the absence of a control group for comparison. Similarly, a six-week study comparing whole-grain wheat sourdough to regular white bread found no difference in blood pressure [41]. In vitro studies have described an anti-hypertensive effect in spelt flours with 96 h of fermentation [42] and quinoa and wheat flours with 46 h of fermentation [43], due to the release of bioactive peptides involved in the inhibition of the Angiotensin-Converting Enzyme (ACE) activity during fermentation, which controls blood pressure by having a vasoconstrictor effect [42,43]. The effects of fermentation vary based on flour type, time, and microorganisms used. Future research could explore genetic variations related to ACE and hypertension to tailor recommendations for sourdough bread consumption. Despite minor variations in blood pressure, we did not observe any additional differences in lipid or glucose metabolism following the 2-month intervention. Additionally, a six-week study found no significant changes in cholesterol or blood sugar between whole-grain wheat sourdough and white bread in adults with normal or high blood sugar levels [41]. Another study with young, healthy adults found that after four weeks of eating “Verna” sourdough or yeast bread, LDL cholesterol decreased, but both types of bread had similar effects on cholesterol and blood sugar, with no added benefits from sourdough [34]. While the long-term benefits of sourdough on blood sugar control are not clear, a meta-analysis suggests it might help lower blood sugar levels after meals compared to other breads [44].

We wanted to determine whether the frequent consumption of sourdough bread could have a long-term effect on satiety, as, to the best of our knowledge, this has not been studied. Nevertheless, in our sample, we did not observe any modifications in satiety-related hormones. Previous studies exploring the acute effects of sourdough on the feeling of satiety have been inconclusive [45,46,47]. Some studies have measured incretins. One study found that the postprandial concentrations of gastric inhibitory peptide (GIP) and GLP-1 were lower after consuming sourdough bread compared to whole wheat and whole wheat barley breads [48]. Another study described a lower ghrelin AUC after consuming einkorn sourdough bread compared to commercial breads. Similar to our study, there were no differences among the consumption of different sourdough breads [49]. There is a significant gap in the study of sourdough bread and satiety, especially among persons with high cardiovascular risk and metabolic syndrome. More studies with well-characterized control groups are needed to understand the short-term effects on satiety and the impact on satiety-related hormones.

Regarding microbiota, the lack of changes in alpha diversity, which indicates species richness and diversity within individual samples, suggests consistent levels of microbial diversity regardless of sourdough bread consumption and its fermentation duration. Similarly, the consistency in beta diversity, which measures the differences in microbial community composition between different samples, indicates no changes in microbial composition across the interventions. These findings are in line with a similar trial on ulcerative colitis patients that investigated a two-month intervention using two different sourdough breads (varying in percentage) [50]. Like our study, this trial reported no differences in microbiota diversity among the 23 subjects. Similarly, a one-week trial with 20 healthy subjects comparing sourdough to white bread found no differences in alpha and beta diversity or phylum-level relative abundances [51]. Despite not finding differences in diversity, we identified some changes in the abundances of certain phyla and families. The Synergistota phylum decreased following sourdough bread consumption, although its abundance remained below 1%. This poorly characterized phylum inhabits human soft tissues and the gut, with a role in amino acid degradation [52]. At the family level, EBLong consumption reduced Enterobacteriaceae abundance. This family thrives in inflamed environments and is associated with inflammatory bowel disease, obesity, colorectal cancer, celiac disease, and antibiotic use [53]. EBLong may reduce conditions favoring Enterobacteriaceae, thereby lowering their abundance. Further studies could explore these changes and their potential effects on gastrointestinal symptoms after sourdough consumption. For EBShort, we observed a decrease in Oscillospiraceae abundance. The genus Oscillospira within this family is linked to leanness, metabolic health, and reduced inflammation [54]. Additionally, we found an increase in Prevotellaceae, a family more abundant in individuals with obesity [55]. Our analysis did not extend to the genus level, making it hard to draw specific conclusions about these changes. Furthermore, the weight-related effects of these families were not reflected in our clinical data. These findings suggest that intestinal microbiota show resilience to changes induced by sourdough bread consumption.

Our study has some limitations. First, the COVID-19 pandemic disrupted our clinical trial, leading to significant dropout rates and affecting the balance between intervention groups, particularly regarding BMI. We attempted to mitigate this by adjusting for covariates such as age, sex, BMI, adherence to the Mediterranean diet, and baseline values. Moreover, although we conducted an analysis to adjust for multiple comparisons, our study requires a cautious interpretation of the results. Second, although the sourdough breads used in this study were similar in composition and the nutritional profiles were closely comparable in terms of energy, carbohydrates, total fat, protein, fiber, and sodium content, variations in ingredients could have influenced our findings. Thus, this study lacked a control group with exact composition but different fermentation times. Third, both the intake of bread and the nutritional assessments relied on self-reports, potentially introducing bias. Lastly, the high dropout rate, biological variability, and specific traits of our study population may limit the generalizability of our findings to broader populations.

## 5. Conclusions

The consumption of sourdough bread may offer potential benefits in inflammation markers in individuals with metabolic syndrome. The gut microbiota, however, did not exhibit differences when comparing both interventions, suggesting that it remains stable despite changes in sourdough consumption. Apart from a higher abundance of Enterobacteriaceae in the group that consumed EBL, no further modifications were observed with a bread with longer fermentation times. More studies with larger samples comparing different fermented sourdough breads and control breads are needed to verify our results and to fully understand the potential benefits and mechanisms of action of sourdough bread in metabolic and cardiovascular health.

## Figures and Tables

**Figure 1 nutrients-16-02380-f001:**
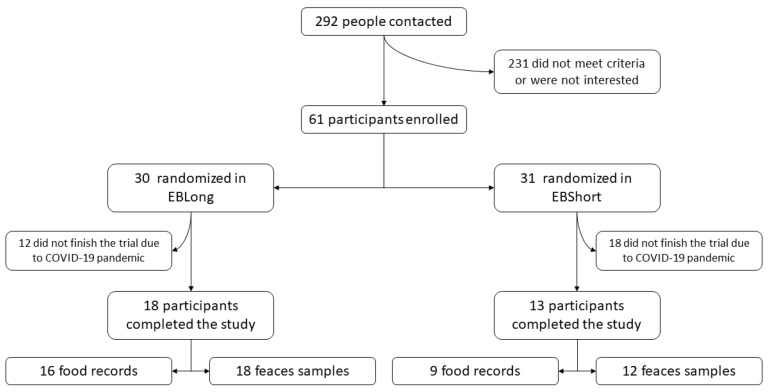
Study flowchart. EBLong, Elias Boulanger^®^ long-fermentation bread; EBShort, Elias Boulanger^®^ short-fermentation bread.

**Figure 2 nutrients-16-02380-f002:**
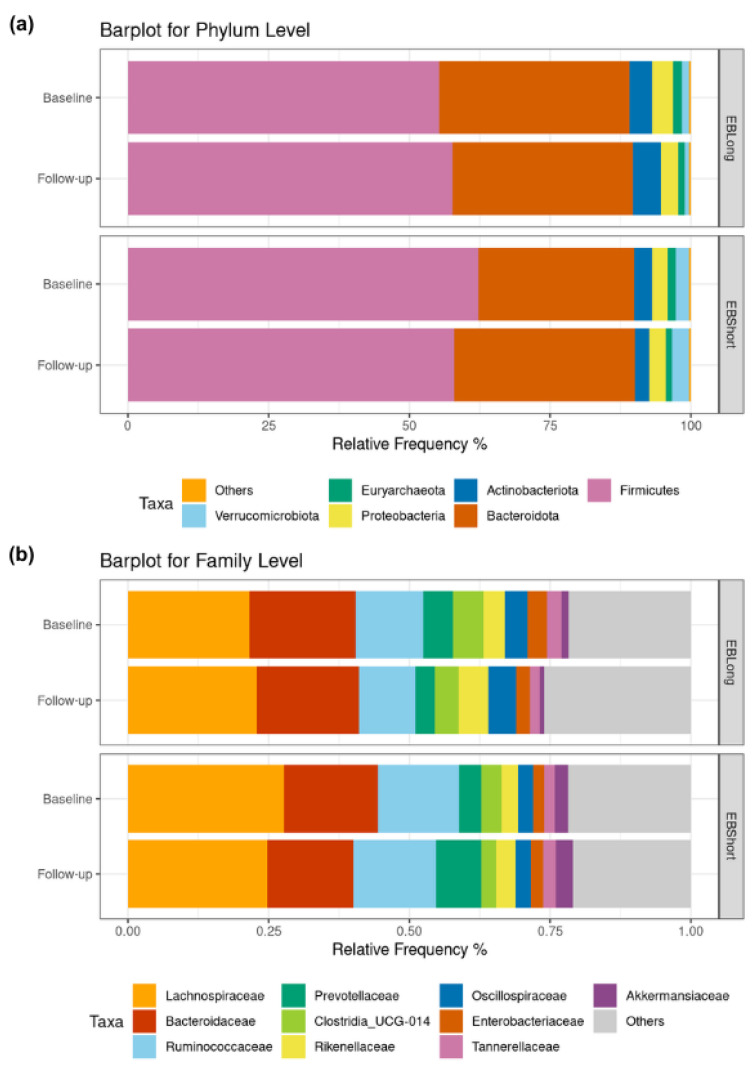
Barplots for relative abundance. (**a**) Displays abundance at phylum level for each intervention. Phyla with relative abundance lower than 1% grouped as “Others”. (**b**) Displays abundance at family level of 10 most abundant taxa.

**Table 1 nutrients-16-02380-t001:** Basal characteristics of participants.

	All	EBLong	EBShort	*p* Value
*n*	31	18	13	
Age, mean (SD)	66.7 (5.94)	66.6 (7.04)	66.8 (4.36)	0.954
Sex, Female, n (%)	16 (51.6%)	8 (44.4%)	8 (61.5%)	0.565
Diabetes, n (%)	26 (83.9%)	17 (94.4%)	9 (69.2%)	0.134
Hypertension, n (%)	30 (96.8%)	18 (100%)	12 (92.3%)	0.419
Triglycerides, mg/dL, median [1st–3rd quartile]	142 [90.5; 174]	146 [90.0; 168]	139 [92.0; 175]	0.889
HDLc, mg/dL, mean (SD)	49.0 (11.7)	50.3 (11.8)	47.2 (11.7)	0.475
BMI, kg/m^2^, mean (SD)	32.8 (3.26)	34.1 (2.84)	31.2 (3.13)	0.015
Scholarity				0.895
Elementary School	13 (43.3%)	7 (38.9%)	6 (50.0%)	
Middle school	9 (30.0%)	6 (33.3%)	3 (25.0%)	
Higher education	8 (26.7%)	5 (27.8%)	3 (25.0%)	
Smoking habit				0.634
Non-smoker	13 (41.9%)	9 (50.0%)	4 (30.8%)	
Smoker	5 (16.1%)	2 (11.1%)	3 (23.1%)	
Former smoker	13 (41.9%)	7 (38.9%)	6 (46.2%)	
Adherence to MedDiet (14 pt), points, mean (SD)	9.71 (2.18)	9.83 (2.07)	9.54 (2.40)	0.724
Basal intake, kcal, mean (SD)	1558 (345)	1552 (387)	1567 (291)	0.900
Physical activity, Mets/day, mean (SD)	2502 (1885)	2320 (1632)	2753 (2234)	0.560

EBLong, Elias Boulanger^®^ long-fermentation bread; EBShort, Elias Boulanger^®^ short-fermentation bread; BMI, body mass index; SD, standard deviation; HDLc, high-density lipoprotein cholesterol; MedDiet, Mediterranean diet.

**Table 2 nutrients-16-02380-t002:** Intragroup differences between baseline and two-month follow-up.

		EBLong			EBShort			All	
	Baseline	Follow-Up	*p* Value	Baseline	Follow-Up	*p* Value	Baseline	Follow-Up	*p* Value
Clinical parameters									
Weight, kg	92.9 (14.7)	93.4 (14.3)	0.710	84 (10.2)	83.5 (9.5)	0.415	89.1 (13.5)	89.1 (13.2)	0.462
Waist, cm	119 (17.3)	114 (12.5)	0.237	111 (10.3)	110 (9.18)	0.796	115 (15.1)	112 (11.2)	0.223
Systolic pressure, mmHg	136 (11.3)	132 (14.5)	0.470	134 (10.1)	135 (9.91)	0.395	135 (10.7)	134 (12.6)	0.830
Diastolic pressure, mmHg	80.2 (12.2)	72.5 (10.2)	0.020	77.6 (12.7)	72.7 (11.3)	0.208	79.1 (12.2)	72.6 (10.5)	0.008
Glucose, mg/dL	125 (30.7)	128 (33.2)	0.162	117 (23.8)	117 (21.9)	0.967	122 (27.8)	124 (29.2)	0.318
Homa Index	18.2 (7.97)	17.3 (7.43)	0.286	21.2 (15.6)	21 (13.8)	0.890	19.5 (11.7)	18.8 (10.5)	0.431
Triglycerides, mg/dL	146 [90; 168]	130 [91; 158]	0.862	139 [92; 175]	124 [84; 179]	0.839	139 [92; 175]	124 [84; 179]	0.814
Total cholesterol, mg/dL	199 (39.5)	202 (38.6)	0.378	189 (38.9)	192 (56.3)	0.707	195 (39)	198 (46.3)	0.444
HDLc, mg/dL	50.3 (11.8)	50.6 (11.7)	0.736	47.2 (11.7)	48.7 (15.4)	0.263	49 (11.7)	49.8 (13.1)	0.282
LDLc, mg/dL	120 (28.6)	125 (35.1)	0.123	115 (33.1)	115 (44.2)	0.918	118 (30.2)	121 (38.8)	0.314
Inflammatory biomarkers									
IL6, pg/mL	2.4 (1.73)	3.06 (1.93)	0.106	2.5 (1.5)	2.14 (1.08)	0.350	2.44 (1.62)	2.67 (1.67)	0.426
IL8, pg/mL	4.49 (2.21)	3.86 (2.02)	0.116	4.63 (2.78)	4.4 (2.04)	0.563	4.55 (2.42)	4.09 (2.02)	0.099
TNF-α, pg/mL	29.6 (9.59)	29.9 (11.8)	0.898	40.7 (15.4)	35.5 (12.4)	0.032	34.2 (13.3)	32.2 (12.2)	0.246
PAI-1, pg/mL	2740 (1070)	2840 (999)	0.466	2750 (529)	2330 (773)	0.018	2740 (872)	2630 (933)	0.318
sICAM, pg/mL	179,000 (67,500)	170,000 (41,800)	0.325	192,000 (59,300)	160,000 (39,200)	0.013	184,000 (63,500)	166,000 (40,300)	0.014
LBP, ng/mL	15,100 (2630)	16,500 (4370)	0.095	14,200 (3820)	13,900 (3690)	0.761	14,700 (3160)	15,400 (4230)	0.259
Satiety-related hormones									
Insulin, pg/mL	423 (201)	388 (168)	0.067	484 (282)	490 (287)	0.797	449 (236)	431 (227)	0.241
Glucagon, pg/mL	520 (188)	493 (177)	0.143	541 (117)	539 (180)	0.949	529 (160)	512 (177)	0.376
GLP-1, pg/mL	164 (97.8)	165 (111)	0.960	187 (125)	223 (122)	0.191	174 (109)	189 (118)	0.304
Visfatin, pg/mL	1910 (1310)	1730 (1380)	0.133	2030 (1440)	1990 (1400)	0.887	1960 (1340)	1840 (1370)	0.364
Resistin, pg/mL	4320 (1720)	4360 (1310)	0.883	6260 (3020)	5630 (2240)	0.339	5130 (2510)	4890 (1840)	0.445
C-peptide, pg/mL	1100 (423)	1050 (358)	0.389	1190 (511)	1190 (613)	0.940	1140 (457)	1110 (481)	0.618
Ghrelin, pg/mL	902 (297)	904 (274)	0.952	1180 (777)	1130 (622)	0.444	1020 (558)	998 (458)	0.531
Leptin, pg/mL	8920 (5110)	8540 (5430)	0.434	9170 (4680)	9120 (4910)	0.897	9020 (4850)	8780 (5140)	0.451

Baseline and follow-up values are presented as mean (SD) or median [1st–3rd quartile]. EBLong, Elias Boulanger^®^ long-fermentation bread; EBShort, Elias Boulanger^®^ short-fermentation bread; HDLc, high-density lipoprotein cholesterol; LDLc, low-density lipoprotein cholesterol; IL6, interleukin 6; Il8, interleukin 8; TNF-α, tumor necrosis factor alpha; PAI-1, plasminogen activator inhibitor-1; sICAM, soluble intercellular adhesion molecule; LBP, lipopolysaccharide binding protein; GLP-1, glycoprotein 1.

**Table 3 nutrients-16-02380-t003:** Intergroup differences between follow-up values.

	EBLong vs. EBShort			
	Non-Adjusted (Diff. [95% CI])	*p* Value	Adjusted (Diff. [95% CI])	*p* Value
Clinical parameters				
Weight, kg	9.82 [0.47; 19.2]	0.050	−0.2 [−2.03; 1.62]	0.829
Waist, cm	4.49 [−3.89; 12.9]	0.303	−4.46 [−9.22; 0.3]	0.082
Systolic pressure, mmHg	−3.16 [−12.7; 6.38]	0.522	−11.6 [−21.1; −2.12]	0.026
Diastolic pressure, mmHg	−0.18 [−8.17; 7.81]	0.966	−6.43 [−14.6; 1.76]	0.140
Glucose, mg/dL	11.3 [−9.48; 32]	0.296	5.71 [−4.76; 16.2]	0.296
Homa Index	−3.64 [−11.2; 3.9]	0.352	0.31 [−3.05; 3.67]	0.858
Triglycerides, mg/dL	−9.53 [−53.2; 34.1]	0.672	−33.3 [−66.6; −0.086]	0.062
Total cholesterol, mg/dL	10.2 [−23.2; 43.5]	0.554	−4.44 [−25.3; 16.4]	0.681
HDLc, mg/dL	1.87 [−7.65; 11.4]	0.703	−1.49 [−4.9; 1.91]	0.399
LDL cholesterol, mg/dL	10.2 [−17.7; 38.1]	0.479	2.03 [−16.1; 20.2]	0.829
Inflammatory biomarkers				
IL6, pg/mL	0.92 [−0.25; 2.08]	0.134	1 [−0.17; 2.16]	0.107
IL8, pg/mL	−0.55 [−1.99; 0.9]	0.466	−0.62 [−1.57; 0.34]	0.219
TNF-α, pg/mL	−5.59 [−14.2; 3]	0.212	3.81 [−3.15; 10.8]	0.295
PAI-1, pg/mL	516 [−135; 1170]	0.131	744 [282; 1210]	0.004
sICAM, pg/mL	9530 [−19,500; 38,600]	0.525	22,100 [2250; 42,000]	0.040
LBP, ng/mL	2520 [−411; 5450]	0.103	1710 [−1210; 4630]	0.263
Satiety-related hormones				
Insulin, pg/mL	−102 [−262; 58.8]	0.224	−22 [−91.9; 47.9]	0.543
Glucagon, pg/mL	−46.3 [−173; 80.7]	0.480	−2.05 [−96.7; 92.6]	0.966
GLP-1, pg/mL	−58.6 [−141; 24]	0.175	−16.9 [−88.1; 54.2]	0.646
Visfatin, pg/mL	−258 [−1250; 731]	0.613	19.9 [−626; 665]	0.952
Resistin, pg/mL	−1270 [−2530; −22.4]	0.056	−5.84 [−1190; 1180]	0.992
C-peptide, pg/mL	−149 [−498; 200]	0.411	50.4 [−205; 306]	0.703
Ghrelin, pg/mL	−225 [−548; 97.1]	0.181	−45.4 [−163; 72.1]	0.457
Leptin, pg/mL	−584 [−4310; 3140]	0.761	−276 [−1920; 1370]	0.745

Intergroup comparisons in follow-up values relative to EBShort were estimated by multivariable linear regression adjusted for baseline values, age, sex, BMI, and MedDiet adherence (14 pt). EBLong, Elias Boulanger^®^ long-fermentation bread; EBShort, Elias Boulanger^®^ short-fermentation bread; HDLc, high-density lipoprotein cholesterol; LDLc, low-density lipoprotein cholesterol; GLP-1, glycoprotein 1; IL6, interleukin 6; Il8, interleukin 8; TNF-α, tumor necrosis factor alpha; PAI-1, plasminogen activator inhibitor-1; sICAM, soluble intercellular adhesion molecule; LBP, lipopolysaccharide binding protein.

## Data Availability

The generation and analysis of the data sets within this study are not projected to be open to access beyond the core research group. This is because the participants’ consent forms and ethical approval did not include provisions for public accessibility. However, we follow a controlled data-sharing collaboration model, as the informed consent documents signed by the participants allow for regulated collaboration with other researchers for study-related research. The data described in the manuscript, alongside the codebook and analytic code, will be available upon request. Researchers interested in this study can reach out to the corresponding author, Montse Fitó (mfito@researchmar.net).

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
