# Peer review of "Sourdough Bread with Different Fermentation Times: A Randomized Clinical Trial in Subjects with Metabolic Syndrome"

_nutrients, 2024, doi:10.3390/nu16152380_

Round 1

Reviewer 1 Report

Comments and Suggestions for Authors

Dear Authors,

I appreciated your work, but in my opinion using two  different flour is a fault too great. In addition, the EBS bread contains many food additives. These 2 breads are not comparable. If you want to compare the effect of short or long fermentation, ingredients must be equivalent

the article is well written

Minor comments:

line 31 You cannot use a number to start a sentence after the dot

line 66 in agreement with last research ( https://pubmed.ncbi.nlm.nih.gov/38157987/ ) it's not correct including wine in a healthy med diet

line 76 please specify that only some species survive to baking

line 363 please correct underlay

Comments on the Quality of English Language

English language is fundamentally correct

Author Response

Thank you for your careful review of the manuscript. Please see the attachment.

Reviewer 2 Report

Comments and Suggestions for Authors

This is an interesting article on a human intervention study aimed at evaluating the possible effect of two types of naturally leavened bread (long and short fermentation) on various health parameters and on plasma biomarkers linked to lipid metabolism, inflammation and satiety, and on the richness and abundance of the intestinal microbiota in patients with metabolic syndrome. Overall, the results are clear, well presented and discussed.

As admitted by the authors themselves, the main limit of the study consists in the low number of subjects who completed the intervention, as a consequence of the high number of subjects who abandoned the intervention. This is probably the reason responsible for the null effects observed, i.e. most of the parameters studied were not altered by the dietary intervention.

Some minor points that should be corrected in the manuscript are as follows:

-Table 1 is never mentioned in the text. Please add a section in the results.

-Figure 2 needs better resolution, as it is difficult to read.

Author Response

Thank you very much for reviewing our paper and for the valuable comments. Please see the attachment.

Reviewer 3 Report

Comments and Suggestions for Authors

The revised manuscript, entitled “Sourdough bread and metabolic risk factors. A randomized,  controlled trial in subjects with metabolic syndrome.” by Karla-Alejandra Pérez-Vega et al.  is  interesting and generally well written but suffers from some  flaws:

The main problem is the low sample size. The initial protocol assumed testing 61 people (31 were randomly assigned to the EBLong intervention and 30 to the EBShort intervention), but finally 31 individuals were included in the study (13 participants in EBShort and 18 in EBLong group). How sample size was determined?

Although the authors used statistical methods to correct for the effect of a small sample, the results should be interpreted with caution.

Laboratory analysis: Have laboratory analyzes been performed in the same run or different runs?

Please include intra and inter -assay CV for laboratory tests of inflammation markers, vascular related hormones and LBP

Please discuss your results in the context of biological variability for selected parameters, e.g. PAI-1

(https://biologicalvariation.eu/)

Please calculate p values for differences in categorical variables (table 1).

Comments on the Quality of English Language

Minor editing of English language required

Author Response

Thank you for your valuable comments. Please see the attachment.

Reviewer 4 Report

Comments and Suggestions for Authors

The submitted manuscript focuses on an important and intriguing topic. However, there are notable concerns throughout various sections of the manuscript and in its overall presentation. This review suggests that substantial revisions would greatly enhance the quality of the manuscript. Specifically, improvements are needed in clarifying the research methodology, rigor in data analysis and interpretation, and coherence between the study’s objectives, results, and conclusions. Furthermore, enhancing the discussion to better present the findings within the existing literature and addressing limitations more explicitly would strengthen the scientific contribution of the manuscript. Please see below for the details:

1)      Title: The title of the paper is, “Sourdough bread and metabolic risk factors. A randomized controlled trial in subjects with metabolic syndrome.” This title may give the false impression that the study used a randomized controlled trial design to investigate the effects of sourdough bread on metabolic risk factors, including a control group not consuming sourdough bread. In fact, this study focused on the effects of two sourdough breads with different fermentation times on clinical parameters, inflammatory biomarkers, and satiety hormones. Please modify the title to clearly reflect the contents of this study.

2)      Abstract: The Abstract lacks specific information on the key findings that form the basis of the study’s conclusions. Additionally, the expression used by the authors “High-quality sourdough bread may offer mild benefits in blood pressure and inflammation for those with metabolic syndrome” overstates the significance of the findings presented in this study.

3)      Introduction: In the Introduction, the authors mentioned, “It has yet to be established whether sourdough has a beneficial impact on health and whether a wholegrain sourdough, combining probiotics and prebiotics, could promote bacteria survival and microbiota equilibrium" (L73-76). However, the authors described the purpose of this study differently, as stated above. Therefore, the Introduction should be revised to align with the study’s purpose and provide a clear rationale for the current investigation.

4)      Materials and Methods:

a) Adding a couple of relevant references to the text under “Bread composition and fermentation process” might be beneficial for potential readers.

b) L225-229: Please confirm the accuracy and completeness of the information regarding the investigation of differences in the follow-up values in the EBLong group relative to the EBShort group using multivariable linear regressions. Also, please see the comment on Table 3 below.

c) L230-234: The description of the application of the Bonferroni correction appears to be inappropriate. The authors need to confirm the consideration of a p-value of <0.017 as the cut-off for statistical significance (also, please review Tables 2 and 3 for this).

d) L235-241: The authors calculated the relative abundances of microbiota at phylum and family levels. Please clarify at which taxonomic level(s) the analyses on alpha and beta diversities were performed. It may be better to add information on the similarity/dissimilarity index used in the principal coordinate analysis here instead of in the figure legend. How was the data treated for normal distribution in the beta diversity analysis? No information on the PERMANOVA test results is available in the text.   

5)      Results:

a) Please ensure that the results are presented in a manner that directly addresses the study's hypothesis and aligns with the title and purpose of the study.

b) Please explain the p-values (indicating the comparisons between variables) in the tables, considering the definition provided under Materials and Methods. In the tables, the consideration of a p-value of <0.025 as significant is not understandable.

c) Table 3: Please check the accuracy and appropriateness of the information used in the footnote under Table 3.

d) L303-308: Please clearly state if the differences between groups were not statistically significant. Otherwise, the relevant discussion may confuse readers regarding the interpretation of these study findings.

6)      Discussion:

a)      The initial part of the discussion is redundant, as it repeats results already presented in the Results section. To enhance readability and relevance, the discussion should be more concise and focused on interpreting the study’s findings.

b)      The authors noted that “after two months of consuming sourdough bread, there was a decrease in diastolic pressure” (L357-358). However, this study did not include a control or comparison group for this purpose. Furthermore, while a paragraph in the discussion addresses satiety-related variables (382-394), the authors did not emphasize this aspect in the results section. The same observation applies to other variables related to metabolic syndrome, which is the primary focus of this study. It is crucial that the discussion aligns with the study’s title, purpose, and results.

c)      The dropout rate is substantial, with 40% of the EBLong group and approximately 60% of the EBShort group failing to complete the study. Such a high dropout rate introduces potential bias and may affect the generalizability of the results. Please explicitly acknowledge this limitation in the interpretation of the study’s findings.

d)      Based on this review, the results do not sufficiently support the conclusions drawn in this study.

7)      English and Scientific Writing: An important concern is the English language and scientific writing throughout the entire manuscript, making it difficult for the reviewer to specify particular changes. The writing contains grammatical errors and lacks clarity. The manuscript would benefit from thorough editing to improve the quality of the English language and scientific writing. Significant revisions are needed to enhance the clarity, coherence, and scientific rigor of the manuscript.

Comments on the Quality of English Language

Please see above.

Author Response

Thank you very much for taking the time to review our manuscript. We believe that your recommendations have improved the paper. Please see the attached document for our responses to your comments.

Round 2

Reviewer 1 Report

Comments and Suggestions for Authors

The Authors correctly answer to my suggestions, adding the main concern to limitations section.

The paper can be published in the present form

Author Response

Thank you very much for your valuable comments and expertise in helping to improve our work.

Reviewer 3 Report

Comments and Suggestions for Authors

Thank you for correcting the manuscript in line with my comments and for the explanations in the cover letter. 

I have a few additional minor comments:

Please provide a statistical tool to calculate the sample size and additionally calculate the power of the test (e.g. for PAI-1) with a sample size of 31 (13 participants in EBShort and 18 in EBLong group). 

Table 2: PAI-1;250? 

Comments on the Quality of English Language

Minor editing of English language required

Author Response

Thank you very much for your review and your valuable contributions. You will find the responses in the document attached. 

Reviewer 4 Report

Comments and Suggestions for Authors

This reviewer would like to thank the authors for their sincere efforts to improve the content and quality of the manuscript. However, due to the statistical issue and other discrepancies mentioned below, the manuscript cannot be accepted in its current format. Please address these issues detailed below.

1. The line numbers referenced by the authors in their response letter and those in the revised manuscript do not match. This discrepancy made it difficult for the reviewer to locate and verify the modifications made by the authors. I kindly suggest that the authors to ensure consistency in line numbering to facilitate the review process.

2. Under ‘Statistical analysis’, the authors mentioned, “We identified three significant factors in our data. We used these factors to apply the Bonferroni correction, thus considering values significant when the p-value was less than 0.05/3 (< 0.017)”.

The phrase “three significant factors in our data” lacks specificity and clarity. It is important to note that the Bonferroni correction should be applied to adjust the significance threshold when multiple comparisons are made, ensuring control over the family-wise error rate. Please review and ensure that the Bonferroni correction is appropriately applied to all comparisons in this study to maintain statistical rigor.

The authors may avoid repeating information about the p-value threshold for statistical significance if it has already been detailed in the 'Statistical analysis' section.

3. It appears that there were some oversights in the manuscript preparation, as the authors acknowledged typos in the previously submitted version, which also continues to persist in the revised manuscript. For instance, in Table 1, the authors stated that 51.6% of participants were female, whereas the abstract reports this figure as 55.7%. Ensuring the accuracy of all information presented in the manuscript is crucial, and I kindly request the authors to address any such discrepancies.

4. The journal uses the term ‘Supplementary Materials’ (not ‘supplemental’).

https://www.mdpi.com/journal/nutrients/instructions (under Back Matter)

Could you please ensure that this terminology is used consistently throughout both the main text and the supplementary materials, as per the journal’s guidelines?

5. The authors are encouraged to carefully review the revised manuscript for improvements in English and scientific writing to enhance clarity, quality, and readability. Here are some typical examples:

Line 38:  Instead of “consume over two-months”, consider “over a two-month period”.

Lines 42-43: Instead of “...mean age: 67...”, consider “...mean age, 67 years; ...”.

Lines 60-62: The whole sentence beginning with “However, a more holistic strategy...” appears vague and lacks coherence. It would benefit from clarification, particularly regarding what is being prevented.

Furthermore, if possible, I recommend having the entire manuscript proofread by a native English speaker.

6. L86: Please delete the word ‘controlled’.

Comments on the Quality of English Language

Please refer to the relevant comment above for the details.

Author Response

Thank you very much for your constructive suggestions. In the attached document, you will find our responses.
